Future stem cell analysis: progress and challenges towards state-of-the art approaches in automated cells analysis

Mohamad Zamani Nurul Syahira 1
http://orcid.org/0000-0001-5808-4348 Wan Zaki Wan Mimi Diyana 1 wmdiyana@ukm.edu.my
Abd Hamid Zariyantey 2
http://orcid.org/0000-0002-7047-1612 Baseri Huddin Aqilah 1
1 Faculty of Engineering and Built Environment, Universiti Kebangsaan Malaysia, Department of Electrical, Electronic and Systems Engineering , UKM Bangi, Selangor , Malaysia
2 Faculty of Health Sciences, Universiti Kebangsaan Malaysia, Biomedical Science Programme and Centre for Diagnostic, Therapeutic and Investigative Science , Kuala Lumpur, W. P. Kuala Lumpur , Malaysia
Karakülah Gökhan
Electronic publication date: 2022 Dec 21
Publication date: 2022
Volume: 10
Electronic Location ID: e14513
Received 2022 Sep 22; Accepted 2022 Nov 14
Copyright: © 2022 Mohamad Zamani et al.
Copyright year: 2022
Copyright holder: Mohamad Zamani et al.
License: This is an open access article distributed under the terms of the Creative Commons Attribution License, which permits unrestricted use, distribution, reproduction and adaptation in any medium and for any purpose provided that it is properly attributed. For attribution, the original author(s), title, publication source (PeerJ) and either DOI or URL of the article must be cited.
License URL: https://creativecommons.org/licenses/by/4.0/

Keywords: Machine learning, Image processing, Artificial intelligence, Microscopic images

Funding: Research University Grant from Universiti Kebangsaan Malaysia GUP-2019-055 This research work was funded under a Research University Grant from Universiti Kebangsaan Malaysia with grant number GUP-2019-055. The funders had no role in study design, data collection and analysis, decision to publish, or preparation of the manuscript.

==============================
Background and Aims

A microscopic image has been used in cell analysis for cell type identification and classification, cell counting and cell size measurement. Most previous research works are tedious, including detailed understanding and time-consuming. The scientists and researchers are seeking modern and automatic cell analysis approaches in line with the current in-demand technology.

Objectives

This article provides a brief overview of a general cell and specific stem cell analysis approaches from the history of cell discovery up to the state-of-the-art approaches.

Methodology

A content description of the literature study has been surveyed from specific manuscript databases using three review methods: manuscript identification, screening, and inclusion. This review methodology is based on Prism guidelines in searching for originality and novelty in studies concerning cell analysis.

Results

By analysing generic cell and specific stem cell analysis approaches, current technology offers tremendous potential in assisting medical experts in performing cell analysis using a method that is less laborious, cost-effective, and reduces error rates.

Conclusion

This review uncovers potential research gaps concerning generic cell and specific stem cell analysis. Thus, it could be a reference for developing automated cells analysis approaches using current technology such as artificial intelligence and deep learning.

Introduction

The etymology term ‘cell’ was coined in the scientific study using a microscope as early as 1665. Over two centuries have passed, and the term stem cell arose in 1868 to describe an organism’s evolvement of a fertilised egg. Numerous laboratories study cells incorporating cells analysis has been conducted over the 19th and 20th centuries until the World War II incident. This incident contributed to the first stem cell transplantation of bone marrow and hematopoietic cell discovery (Mathe et al., 1959) and repeated experiment works (Gregory et al., 1968; Jacobson et al., 1949; Jacobson et al., 1951). Stem cells are mainly categorised into embryonic stem cells (ESCs), adult stem cells, and induced pluripotent stem cells (iPSCs), which can be further classified according to differentiation potency known as totipotent, pluripotent, multipotent, oligopotent, and unipotent. Stem cells generate tissues through their differentiation potency, forming mature and functioning cells while retaining stem cells’ pool size via self-renewing ability. Adult stem cells and iPSCs play a greater role in medicine for the regeneration and repair of damaged tissue than ESCs, which are hampered by ethical issues (Seita & Weissman, 2010).

To date, hematopoietic stem cells (HSCs) are among the best-characterised adult stem cells with known multipotency properties (Lin, Otsu & Nakauchi, 2013). Having a remarkable regenerative potential, HSCs have been widely and successfully used for therapeutic purposes via bone marrow and HSCs transplants (Hamid et al., 2020), a standardised stem cell therapy for haematological diseases such as aplastic anaemia, leukaemia, and immunodeficient disorders. Apart from HSCs, iPSCs utilisation also confers excellent therapeutic potential, particularly in regenerative medicine (Yuasa & Fukuda, 2008). Cell-based assays using iPSCs have been used for new drug development and toxicity screening. Since iPSCs possess similar stem cells potency as acquired by embryonic stem cells, they are becoming a favourable source of stem cells being used in clinical testing for drug screening, disease modelling, and cell-based therapy (Fakunle & Loring, 2012; Tanaka et al., 2014; Avior, Sagi & Benvenisty, 2016; Kusumoto, Yuasa & Fukuda, 2022).

Before application in research and stem cell-based therapy for regenerative medicine, stem cells’ potency and function must be identified and confirmed, often requiring a trained scientist or operator to conduct the analysis. The requirement for microscopic observation and manual analysis of stem cell characteristics is fundamental in specific analytical procedures. However, these procedures are mainly performed using conventional methods (Ramakrishna et al., 2020). For example, functional characterisation analysis of HSCs and iPSCs requires microscopic identification and classification of colony subtype via colony-forming assay. A trained operator must perform this conventional approach; laborious, time-consuming, and prone to errors in data reporting. Hence, this review study is intended for clinicians, scientists, and researchers recently enlightened about automated systems to analyse generic cell and stem cells, particularly in HSCs, iPSCs, and other cell types. This automated system could foster and empower stem cell research and its application for therapy.

Therefore, an introduction to this review study of the generic cell and stem cell analysis is briefly explained in this article. In this review, we present a literature study based on methodology concerning article selection as described in “Survey Methodology”. In “An Overview of Cells and Stem Cells”, we provide an overview of the beginning of cell and stem cells, followed by a brief explanation regarding cell analysis evolution back in the 16th century towards the state-of-the-art analysis approaches in the 21st century, including bacterial colonies in microscopic images in “Evolution of Cell Analysis Approaches”. Then, fully developed cell colonies and stem cell analysis system will be thoroughly explained in “Challenges in Cell Analysis Approaches”. Finally, a summary of the overview of stem cells and the execution of automated analysis using deep learning will be the concluding remarks of this review, as disclosed in “Conclusions”.

Survey methodology

A literature survey on a particular subject was performed to filter and review the evolution of various approaches for general cell analysis. In this study, we aim to understand and systemise the reviewed publications from the top list of academic search engines, which are Google Scholar and selected publisher databases with citation index traced by SCIMAGO journal ranking. For example, Elsevier, IEEE Xplore, Web of Science, and several primal publishers such as JSTOR for certain manuscripts, including Dutch manuscripts, for originality and novelty concerning cell analysis approaches. For the primal publisher, some ancient manuscripts require extra effort to understand the author’s lengthy and indirect explanations. Therefore, time period restrictions were applied for historical evolution studies from the 16th century to the current evolution study in the 21st century. By referring to guidelines to systemise review, such as the origin of stem cells (Ramalho-Santos & Willenbring, 2007), microscope inventions (Hajdu, 2002) and cell analysis evolution (Mitra-Kaushik et al., 2021), including technical journals for advanced cell analysis. This part consists of various types of cell analysis, such as traditional (manual and clinical practice), conventional (image processing and machine learning), and state-of-the-art (deep learning).

The publications were selected according to the inclusion and exclusion criteria of the most appropriate manuscripts from the mentioned databases. Three inclusion criterions were involved in this review, starting with the identification phase, followed by screening and inclusion phase. A total of 183 articles were selected and identified with appropriate title manuscripts contributed to cell analysis. Then, the screening phase is based on the contents of the abstract and keywords such as “microscope”, “stem cell”, “single-cell analysis”, “cell identification”, “cell counting”, “image processing”, “machine learning” and “deep learning”. Roughly 65 of 183 articles were sorted out, and about 118 were sorted in for downstream data extraction. Finally, 101 articles were extracted for references throughout this review study. The review articles in the cell analysis evolutions section and the research articles without complete results were excluded for exclusion. To be noted, Mendeley has been used as a citation software manager. Figure 1 illustrates the article’s inclusion and exclusion process for this review based on Prisma guidelines (Abdolrasol et al., 2021; Miah et al., 2022).

Figure 1 A block diagram of review methodology for this study.

An overview of cells and stem cells

In the late 1950s and early 1960s, a multidisciplinary work between two scientists, a cellular biologist and a biophysicist, Ernest McCulloch and James Till, conducted scientific research on the susceptibility of mice bone marrow tissue to radiation exposure (McCulloch & Till, 1960). From nuclear accidents, scientists acknowledge and theorise that there is something in the blood that can treat cancer or radiated cells from healthy cells. Both scientists injected healthy mice bone marrow cell transplantation into irradiated mice. Afterwards, they noticed small lumps representing a colony appearing on blood filter cells called the spleen of mice (Till & Mcculloch, 1961). The appearance of small lumps is known as hematopoietic stem cells (HSCs). This experiment has been conducted repeatedly to understand the sensitivity of healthy cells in mice bone marrow to radiation (McCulloch & Till, 1962) and cytological evidence between both cells (Wu et al., 1968). Through repeated experiments, scientists discovered stem cells from spleen colonies in bone marrow, which shows that a single cell can produce colonies by direct cytology (Till & Mcculloch, 1961; McCulloch & Till, 1962; Becker, Mcculloch & Till, 1963; Siminovitch, Mcculloch & Till, 1963). A stem cell has self-renewal ability and differentiates into a new cell categorised into multipotent stem cells. The HSCs are responsible for the differentiation of the blood cells such as erythrocyte (red blood cell), leukocyte (white blood cell), and platelet cell lineages through the hematopoiesis process (Wu et al., 1968; Chow et al., 2021). The hematopoiesis process has been practically demonstrated in nuclear survivors and irradiated mice bone marrow transplants. Therefore, it is very crucial to maintain hematopoiesis with HSC differentiation (Dewi et al., 2020).

Stem cells proliferate through self-renewal division and develop into contrasting or specialised cell types. Self-renewal is the stem cell division process to generate more stem cells, and multi-lineage differentiated progenitors while preserving the undifferentiated state of new stem cells (Worton, Mcculloch & Till, 1969; Weissman, 2000; Kolios & Moodley, 2013; Lin, Otsu & Nakauchi, 2013). Undifferentiated cells define the state of immature cells that have not yet been assigned to specific cells function. The evolutions of stem cells are based on single cell (clonal), where the changing process refers to differentiation and potency to form specific and different cell types and tissue (Coulombel, 2004; Loya, 2014; Łos, Skubis & Ghavami, 2018). The established form of stem cells is embryonic stem cells (ESCs) and adult stem cells (ASCs). Furthermore, stem cells can be further classified based on their differentiation potency and origin. Potency refers to stem cells’ ability to differentiate into different types of cells (have a specialised function). Regarding potency, stem cells are divided into five main categories: totipotent, pluripotent, multipotent, oligopotent, and unipotent (Maleki et al., 2014). The highest hierarchy in differentiation potential of a stem cell is known as totipotency which enables a single stem cell to generate embryo and extra-embryonic tissues (Kolios & Moodley, 2013; Zakrzewski et al., 2019). An example of a totipotent stem cell is a fertilised egg (zygote).

The second stage of stem cell potency is called pluripotent stem cells (PSCs), whereas a form of blastocyst’s inner cells masses from 4 days of zygote formation. Pluripotent stem cells can divide and differentiate into three germ layers: ectoderm, endoderm, and mesoderm (Weissman, 2000; Lin, Otsu & Nakauchi, 2013), all of which are derived from ESCs. Reprogrammed somatic cells, induced pluripotent stem cells (iPSCs), are the first artificially generated stem cells to mimic the function of ESCs (Takahashi & Yamanaka, 2006; Zakrzewski et al., 2019). Since then, iPSCs have been regarded as the popular pluripotent stem cells used in stem cell research and therapy. The multipotent stem cell has restricted differentiation potency compared to ESCs and iPSCs. Mesenchymal stem cells (MSCs) and HSCs are an example of multipotent stem cells due to their ability to differentiate into multi-lineage and multiple types of cells. HSCs can further differentiate to form oligopotent stem cells/progenitors with limited differentiation ability to form cells for specific lineage, such as myeloid and lymphoid (Ebihara et al., 1997; Coulombel, 2004). Finally, fully specialised cells that can reproduce to generate more tissue-specific cells are defined as unipotent stem cells such as skin, cornea, muscle, pancreas, etcetera (Tatullo et al., 2019).

Evolution of cell analysis approaches

This section describes the cell analysis starting with the history, which prescribes the origin of the cell discovery through the invention of microscope observation in the 16th century (Ball, 1966; Hajdu, 2002; Ponti & Muscatello, 2015). The cell analysis approaches evolved using a photoelectronic method by counting cells through flowing fluid, gas and light with a generated electronic pulse (Moldavan, 1934; Coons, Creech & Jones, 1941; Gucker et al., 1947; Guyton, 1946; Gucker & O’Konski, 1949; Crosland-Taylor, 1953; Coulter, 1953; Graham, 2003), and the counted cells can be sorted out according to their cell type using the cell sorter method (Mie, 1908; Twersky, 1964; Fulwyler, 1968; Wyatt, 1969; Bonner et al., 1972; Wyatt et al., 1988). The approaches have progressed as technology emerged in the late 20th and early 21st centuries. Digital image processing (DIP) has paved the way for microscopic image analysis using processing unit capability with the help of camera technology. Nevertheless, the development of technology evolved to meet the high demand requirement, which included the automated approaches of cell analysis using deep learning. Figure 2 illustrates the evolution of the history to the current cell analysis approaches.

Figure 2 Evolution of cell analysis: from the microscope to the deep learning approach.

The camera’s invention from the 11th century until 1966 played an important role in this evolution (Tbakhi & Amr, 2007; Kobayashi, 2002; Klug & De Rosier, 1966).

The history of cell analysis (from 16th century)

Microscope inventions: In the early 1600s, Galileo made a microscope to observe minuscule things representing microorganisms (Haden, 1939, 1942). Later in 1665, Robert Hooke invented a simple single lens of the microscope, as illustrated in Fig. 3. One of Hook’s discoveries was when he examined a thin slice of the cork, a dead tissue found in the bark tree that looked like a honeycomb. Hooke expressed his vision on the first microscopical stoma, where the term ‘cell’ was coined to describe the building blocks of living things, the etymology of the word used in biology. After a decade had passed, Antoni van Leeuwenhoek discovered the first detailed cell description specified the living cells of bacteria and protist parasites as well as the red blood cell as he continued blood capillaries observation by Marcello Malpighi back in the early 1660s (Harris, 1911; Packard, 1928; Saraf & Cockett, 1984; van Leeuwenhoek, 1677; Lane, 2015). Photoelectronic techniques: Moldavan has proposed a photo-electric system for microscopical cell counters (Moldavan, 1934). Early bacteriologists used this method to estimate the number of bacteriophages, enzymes, and viruses. This approach aims to have the photo-electric tool automatically record the presence of each microscopical cell flowing through the capillary tube to amplify the generated micro-current. In the mid of 1950s, a principle of automated cell counting called the Coulter Principle was proposed inspired by Moldavan’s microscopic observation of photo-electric cell counting passing through a capillary (Coulter, 1953). A few years later, Wallace introduced an automatic system of blood cell counters that made cell distribution size faster (Coulter, 1956).

Figure 3 Robert Hooke’s invention and microscope discovery in Micrographia (Hooke, 1665), (A) a microscope and (B) a thin slice of the cork (the origin ‘cell’ term were coined).

Cell sorter techniques

Various modalities have carried out the evolution of cell analysis in the past. The cell sorter technique is generally used to analyse the physical characteristics of cells or particles using flow cytometry and flow sorting method (Fulwyler, 1968; Cram & Arndt-Jovin, 2005). Fluorescence has been widely used in biology for cell stains marker. The quantitative measurement of cell fluorescent emissions light called the cell microfluorometry method (Van Dilla et al., 1969). In 1972, Bonner et al. (1972) introduced a rapid instrument for biological cell sorting analysis called a flow cytometry technique. The cells with different fluorescent compounds are injected into a narrow tube of a flowing stream and lighted up with a laser beam. The fluorescent compounds used in this instrument include fluorescence, quinacrine mustard, and acridine orange (Bonner et al., 1972). Table 1 lists the evolution of cell analysis from a historical point of view from the microscope invention prior to photoelectronic and cell sorter techniques. In summary, this historical review describes the invention and proposed approaches from the origin of single-cell identification in the 16th century until pre-modern cell analysis techniques in the 20th century. By analysing the approaches, the research potential and gap could be determined for future research suggestion improvement.

Table 1 Evolution of the cell analysis history.

Ref.	Approach/Methods	Cell/Particle analysis	Advantages	Disadvantage	
		Detection	Counting	Size measurement	Sorting	
Moldavan (1934)	Cells counting using photo-electric technique.		✓		✓	Capable in microscopic cells counting.

	Faint visibility of microscopic cells.

Need specific requirement tools for better result

	
Coons, Creech & Jones (1941)	The detection of pneumonia bacteria by an ultraviolet-excitable fluorochrome conjugated antibody.	✓				Capable in clear pneumonia bacteria detection.

	Need understanding of fluorescence usage.

	
Guyton (1946)	Electronic counting and size determination of particles.	✓	✓	✓		Capable in particle count and size measurement.

	Particle detection is not smaller than 2.5 µm.

Rejected due to device sensitivity deficit.

	
Gucker et al. (1947)	Particles counter using photoelectronic.	✓	✓	✓		Detect particles down to 0.6 µm.

	Need high understanding of complicated electric circuit.

	
Gucker & O’Konski (1949)		✓	✓	✓		Improvement of particles visibility.

	
Crosland-Taylor (1953)	A device for counting small particles suspended in a fluid through a tube.		✓	✓		The first successful small particles counting in a fluid under microscopical observation with electronic counter alignment.

	High apparatus requirement.

	
Coulter (1953)	Means for counting particles suspended in a fluid.	✓	✓	✓		The first automated cell counting based on Coulter Principle.

	Difficulties to obtain a perfect result.

Requires calibration.

High-cost and complex apparatus requirements.

	
Coulter (1956)	High-speed automatic blood cell counter and a cell size analyser.	✓	✓	✓		Effective in cells detection and counting up to 6,000 individual cells per second automatically.

The minimum function of the oscilloscope and readily understood by average medical laboratories.

	Possibility cell loss count.

The electronic circuit must be correctly connected.

	
Fulwyler (1968)	The first cell sorter.				✓	Able to separate different types of cells.

	Requires understanding of complex circuit and apparatus design.

	
Van Dilla et al. (1969)	Cell microfluorometry: a new method for the rapid measurement of biological cells stained with fluorescent dyes.		✓	✓		Able to measure human leukocytes 104 to 105 cells per minute.

	Apparatus requirement to perform the cell measurement.

	
Bonner et al. (1972)	The first Fluorescence Activated Cell Sorter (FACS) instrument.	✓	✓	✓	✓	High result accuracies.

The result can be obtained in an instant as the process was done.

	Slow stream flowing for FACS sorting.

Expensive technology.

Requires expertise to perform the analysis.

	

Cell analysis using DIP and machine learning approaches (from 19th century)

In a literature study, the naked eye discovered the image of single-cell identification through microscope observation. In other words, the discovery was carried out with eyes vision. Although the inability to analyse the invisible data (visualisation) could be saved in brain memory with the help of the retina through the optic nerve, the illustration can be retrieved through a sketch. This analogy from a pair of biological cameras has been a commence which brings to the beginning of the physical cameras that can capture the image instantly and store it in physical memory.

DIP focuses on improving image representation for machine interpretation with image analysis purposes that led to the start of computer vision until today. Computer vision consists of a digital image processing approach that has performed several machine learning techniques. Machine learning is a subset of artificial intelligence (AI) that enables the programmed system with an algorithm to analyse data from a learning experience (Zamani et al., 2020). Previous studies have paved the way in cell analysis study including detection, counting, cell size measurement, and cell sorting. However, all the experiments have been performed clinically and conducted with expertise. Nonetheless, a scientist has developed physical instruments and devices for practical use. Regardless of this benefit, most of the equipment is expensive and inappropriate for researchers for trial and error. Various cell analysis approaches have been proposed for image detection and classification using image processing and machine learning in the late 20th and early 21st centuries. For example, single smooth muscle cells have been acquired using a developed digital imaging microscope by utilising fluorescence probes (Fay, Carrington & Fogarty, 1989). The acquired three-dimension dataset was analysed with the optic processing reverses distortion to identify the characteristic of an image region. In 2002, an analysis of white blood of various lineage and maturity was performed (Hengen, Spoor & Pandit, 2002). The morphological feature was applied in feature extraction, where the nucleus cell’s structure was analysed using watershed transform and Gabor filter; meanwhile, the cell shape was analysed using rotational invariant contour. Based on these features, the Gaussian-Bayesian classifier was used for classification.

Haematoxylin and eosin-stained (H&E) image gas was automatically identified using two neural network layers with a 10-fold cross-validation technique (Bhagavatula et al., 2010). H&E staining helps in cell identification and provides cell information, such as the shape of cell tissue. The result shows that accuracy classification gives an average of about 74.9% to 93.2% using pixel-based classification. In 2012, Lowry et al. (2012) performed a multi-stage Bayesian level algorithm to segment the iPSC colony. Some of the colonies were misclassified, but apparently, the tested iPSCs have been successfully segmented. Guan et al. (2014) classify human embryonic stem cells (hESCs) using statistical Gabor descriptors and show they classify more accurately than histogram-based features. A stem cell marker has been proposed to detect cancer stem cells in the microscopic image using a CD13 staining marker (Oğuz et al., 2015). Oğuz et al. (2015) applied a support vector machine (SVM) to classify cancer stem cell images and successfully developed CD13 positive cancer stem cells marker on the microscopic image. In 2018, a mesenchymal stem cell (MSCs) classification was performed based on feature extraction using Wavelet and classified using SVM, resulting in greater accuracy than information gain and genetic algorithm techniques (Sreedevi & Pachaiammal, 2018).

Hyperspectral imaging (HSI) microscopy is a 2D image technique used in biomedical imaging for cell analysis. Ogi et al. (2019) have developed a highspeed HSI method to classify neural stem cells with 94% precision using pixel-wise ML. In 2020, a cell classification platform for human PSCs (hPSCs) was proposed for cell-based therapy from the cell’s glycome using a supervised ML approach (Shibata et al., 2020). Shibata et al. (2020) employed linear classification and neural network to classify five classes of annotated lectin microarray data from hPSCs: mesenchymal stromal cells, endometrial cells, endometrial and ovarian cancer cells, cervical cancer cells and pluripotent stem cells. Both supervised ML approaches have achieved 89% and 97% accuracies, respectively, for the hPSCs classification system. Recently, the ML approach has also been implemented to predict allogeneic stem cell donors from the granulocyte colony-stimulating factor (G-CSF) of HSC (Xiang et al., 2022). The established prediction methods were performed by cross-validation (10-fold) with appropriate hyper-parameter using a grid search technique. The best ML prediction models are random forest and AdaBoost with the area under the curve (AUC), 89% and 88% scores, respectively, compared to a decision tree, linear regression, SVM, feedforward neural network and gradient boosting.

Image processing and ML have been widely implemented in microscopic images with benefits based on their reliability and robust technique. However, the accuracies of image analysis depend on the techniques and experiences of the researchers. Furthermore, image processing and machine learning approaches are laborious to find the most significant image features with the appropriate algorithm, which is very time-consuming.

Automated cell analysis using deep learning (21st century)

Previous and current studies have proposed and suggested the appropriate experimental work for image analysation through a deep neural network (DNN). A test analysis of the stem cells in the form of a microscopic image where the type of cells is rationally needs to be validated by the cells specialist (cytobiologist).

An automated assessment of stem cells approach gives rise to DNN in recognition/classification of HSCs and iPSCs colonies, although there is minimal comprehensive study related to these stem cell colonies. In 2017, Kavitha et al. (2017) developed a new vector-based convolutional neural network (V-CNN) to classify feature vectors of healthy and unhealthy colonies by differentiating features of the colony. Before V-CNN classification, the colony undergoes image segmentation and feature measurement to compute morphological and texture features. This segmentation achieved a robust segmentation. However, it is challenging to compute the segmented colony region for the ensuing stem cell characteristics measurement (Paduano et al., 2013). The proposed V-CNN classification performance was then compared to SVM classifier by five-fold cross-validation. The performance of V-CNN achieved high accuracy above 90%, while the SVM classifier had only 77% accuracy. Therefore, V-CNN has outperformed the conventional classifier (SVM), which has a high potential for a reliable framework for automated recognition of iPSCs colonies.

A primary hematopoietic progenitor identification was proposed in the same year by differentiating lineage choice prediction using DNN on image patches from brightfield microscopy and cellular movement (Buggenthin et al., 2017). The hematopoietic progenitor stem cells are differentiated into granulocytic/monocytic lineage (GM) and megakaryocytic/erythroid lineage (MegE). The idea of the CNN combination is to extract the image features automatically with long short-term memory (LSTM) of recurrent neural network (RNN) architecture for cell dynamics models. A proposed CNN was adopted from the LeNet structure with three convolutional layers, a nonlinear activation function: ReLU (alternately after the convolutional layer), a max-pooling layer, and two fully connected layers. The limitation of DNN deployments, such as overfitting during the network training, Buggenthin and the team have overcome the problem by using dropout layers after a fully connected layer where the weight of the unwanted features is discarded. Then, the annotated cells (visible cell marker) and latent cells (non-visible cell marker) are fed into a softmax classifier for classification purposes. For an RNN, the output of the extracted features is transferred to an LSTM with 20 hidden nodes. The CNN-RNN proposed method predicts the three-lineage choice of hematopoietic progenitor with fast and robust for new data as it is applied with a low-level feature calculation. On average, the area under the curve (AUC) performance with 87.3% for annotated cells and 79.3% for latent cells. The CNN implementation has shown a remarkable performance in the morphological identification of iPSCs and haematopoietic cells. Human iPSCs represent an experimental model for cell development. A deep learning CNN architecture was implemented to automatically classify and recognise iPSC regions in microscopy images (Chang et al., 2017). The microscopy image of iPSCs was divided into six classes based on cell patterns. The six regions are with no cells, separate cells, unclustered cells, clustered cells, tightly clustered cells, and regions with iPSCs. Five convolutional layers and two fully connected layers are used for CNN architecture. Overall, the classification of six classes archived Top-1 9.2% and Top-2 0.84% of error rates with average time execution of approximately 0.076 s.

Kusumoto et al. (2018) have proposed identifying endothelial cells derived from iPSCs without immunostaining or lineage trace requirements. To establish an automatic prediction, they implement two DNN architectures: LeNet contains two layers of the convolutional network, max pooling and fully connected, respectively, and AlexNet with five convolutional layers and three max pooling and fully connected layers. A total of 800 images were used, 600 for training and 200 for validation. The network parameter was optimised to increase prediction accuracy and validated by K-fold cross-validation. The identification achieved 70% for both accuracy and F1 score (LeNet) and AlexNet score 90% accuracy and 75% F1 score. Based on the network performance, the network depth and pixel size’s image play an important role in correlating the prediction accuracy. In 2020, a microscopic image of erythrocytes was acquired using mobile microscopy and automatically segmented to identify malaria using a deep residual network (ResNet) (Pattanaik et al., 2020). ResNet is one of the DNNs modified with the adoption of batch normalisation implemented after the convolutional layers works. The modified network called multi-magnification ResNet (MM-ResNet) was fed with multi-magnification of microscopic images of 200×, 400× and 1,000× magnification. The proposed system has achieved slightly low accuracy (0.10%) compared to GoogleNet. Despite that, the proposed system beats the test error rate of GoogleNet with 19.93% compared to 25.86%.

Researchers have recently proposed a classification/identification of bacterial colonies in another type of microorganism with the same culture in microscopic images as HSCs and iPSCs; researchers have recently proposed a classification/identification of bacterial colonies. Early detection and classification of bacterial colonies have been proposed using pseudo-three dimensional (3D) of DenseNet (Wang et al., 2020). The images of live bacteria colonies are captured by a monochromatic complementary metal-oxide-semiconductor (CMOS) image sensor after being incubated for 24 h in four time-consecutive frames in each time-lapse imaging for detection purposes. A total of 6,969 E. coli, 2,613 K. aerogenes, and 6,727 K. pneumoniae are the estimated images used for the training model, and about 965 colonies were tested of the three classes. All the images were augmented with rotation by 90 degrees and flipping randomly. The authors modified the DenseNet architecture by replacing two-dimensional (2D) convolutional layers with three-dimensional convolutional layers. The modified network was then randomly initialised with a hyper-parameter learning rate of 0.0001, batch size of 64, and optimised by an adaptive moment estimation optimiser (Adam). While training the data, the learning rate was decayed by half for every 20 epochs to stabilise the network accuracy. For classification purposes, the images used only contained true classes of about 9,400 growing colonies with the same hyper-parameter setting except for the learning rate decaying during the network training by 0.9 for every 10 epochs. The training process for classification took 10 h more than the detection task, resulting in 80% (classification) and 95% (detection) using the same environment of dual graphic processing units (GPUs) (GTX1080Ti) Nvidia. Despite the small number of training data used in classification resulting in low accuracy, it took more time to process, which proved that the network probably encountered an underfitting problem.

Two years later, another experimental work has been proposed using the same network architecture and bacterial colonies classes as (Wang et al., 2020) DenseNet by replacing the 2D convolutional layer with pseudo-3D convolutional layers except for the hyper-parameter settings. The authors (Li et al., 2022) have demonstrated the first use of a real-time detection system with a thin-film-transistor (TFT)-based image sensor by automatically counting and identifying CFU species using a deep learning approach. Unlike Wang et al. (2020), the TFT-based image sensor is field-portable as there is no requirement for image scanning on an agar plate and time effective as the image was taken within 5 min. A total of 889 colonies of positive class and 159 colonies of the negative class were augmented by rotation and flipping as well as Wang et al. (2020) performed training, and another 442 colonies (positive class) and 135 colonies (negative class) were assigned for testing. This experimental work has performed by 5-fold cross-validation with the hyper-parameter setting of 0.0001 learning rate and batch size of 8, and the network was scheduled to decrease the learning rate by 0.8 for every 10 epochs and optimised by Adam optimiser. As for the classification task, the authors assign 442 colonies which are less than the detection task with the same network architecture but have a contrast for the hyper-parameter setting. Regardless of the less training data used and time-effective, this experimental work has achieved 92.6% sensitivity and 100% specificity (detection), and 100% classification accuracy, which has been improved from previous work in 2020. This achievement shows that despite the total training data and balancing data used, the combination of hyper-parameter selection is vital for converging the network and avoiding overfitting and underfitting problems in deep learning.

Manual work to count the CFUs was laborious, time-consuming, and prone to error. A bacterial colonies detection has been proposed and aimed to automate the CFU cell counting using U-Net architecture for vaccine development (Beznik et al., 2022). The CFUs were grown from the number of viable and proportion of virulent bacterial colonies that were cultured on a blood agar plate. A plate of CFU consists of four classes, namely bgv+ colony (virulent), bgv− colony (avirulent), background, and border from 108 images. From the point of view, the researchers have highlighted the challenges in deep learning apart from millions of parameters and large-scale data requirements; they also encountered the imbalance of data as the petri dish image mostly represented by background class and solved it by altering the loss, and the image noise such as reflections from light-emitting diodes (LEDs). This work implements two networks of regular U-Net (Ronneberger, Fischer & Brox, 2015) with a batch normalisation layer and pre-trained U-Net from ResNet, Inception-ResNet, and DenseNet (Russakovsky et al., 2015) in search for the best hyperparameter combination. These models have been trained using three learning rates (0.001, 0.0001, and 0.00001), batch size of 8, epochs 10, and Adam optimiser. ResNet-152 has outperformed other models for cell counting with learning rate of 0.0001 with precision and recall achievement, bgv+ (90.6%, 92.7%) and bgv− (91.9%, 90.5%), respectively. However, the proposed framework has scored less than 74.5% for true positive class detection, which is quite low for state-of-the-art performance. This result is probably due to the small-scale training data used with the deep layer network such as ResNet and DenseNet, and imbalanced data that leads to the underfitting problem.

Data imbalance is a well-known scenario in medical imaging when training using a deep neural network (Johnson & Khoshgoftaar, 2019; Cano et al., 2021). Zhang et al. (2022) used a balanced total data sample for training and testing to identify three bacteria classes to overcome this problem. A total of 3,059 images of Salmonella Enteritidis, 3,110 images of Salmonella Paratyphoid, and 3,092 images of Salmonella Typhimurium were used in this work. These are the number of surface-enhanced Raman spectroscopies (SERS) spectral samples of bacteria in the water bath. All the SERS samples were fed to seven different architectures. Regardless, the authors did not mention the hyper-parameter used in this experimental work training process for evaluation purposes. Despite that, the stacking-CNN has been well-converged at epochs 15 without experiencing an overfitting problem. This model has outperformed other architectures with the accuracy of 98.41% (Salmonella Enteritidis), 97.89% (Salmonella Paratyphoid), and 98.31% (Salmonella Typhimurium). This work shows that balancing data could help achieve excellent performance despite the combination of hyper-parameter during the training process.

In summary, Fig. 4 illustrates the Venn diagram briefly explaining each field. It shows the relationship between computer vision representing image processing approaches with artificial intelligence that includes machine learning and deep learning in the hierarchy for image data representation in cell analysis approaches. In addition, this review has listed the research gap for future guidelines in Table 2.

Figure 4 A hierarchy of computer vision and artificial intelligence.

Table 2 Current approach of automated cell analysis using deep learning.

Ref.	Method	Cell analysis scope	Research gap	
Identification	Recognition/Detection	Classification	
Kavitha et al. (2017)	A new vector-based convolutional neural network (V-CNN) to classify a feature vector of healthy and unhealthy of iPSCs colonies.

Undergo image segmentation and feature measurement to compute morphological and texture features.

Comparison between the proposed V-CNN and SVM classifier by five-fold cross-validation.

		✓		V-CNN achieved high accuracy above 90% and the SVM classifier only 77%.

V-CNN has outperformed the conventional classifier (SVM).

High potential for reliable framework in automated recognition of iPSCs colonies.

Quite challenging to compute the segmented colony region for the ensuing stem cell characteristics measurement.

	
Buggenthin et al. (2017)	Combination of CNN and LSTM of RNN for identification of hematopoietic progenitor.

CNN adopted from LeNet structure.

Three convolutional layers.
Three max-pooling layers.
Two fully connected layers.
A softmax classifier for classification. For an RNN, the output of the extracted features is transferred to an LSTM with 20 hidden nodes.

		✓		Fast and robust for new data for training.

Low level feature calculation requirement.

However, the score of identification in not stated.

	
Chang et al. (2017)	Six classes microscopy images of iPSCs based on cell patterns.

A region with no cells, separate cells, clustering cells, clustered cells, tightly clustered cells, and a region with iPSCs.

CNN: Five convolutional layers and two fully connected layers

		✓	✓	Top-1 9.2% and Top-2 0.84% of error rates.

The average time execution is 0.076 s approximately.

	
Kusumoto et al. (2018)	Adaption of two DNN architectures for identification of endothelial cells derived from iPSCs

LeNet (two layers of convolutional network, max pooling and fully connected)

AlexNet (five convolutional layers and three layers both max pooling and fully connected layer).

A total of 800 images (600 images for training and 200 validation).

Network parameters are iteratively optimised.

	✓			LeNet 70%:

accuracy and F1 score AlexNet:

90% accuracy
75% F1 score	
Pattanaik et al. (2020)	Modified deep residual network called multi-magnification ResNet (MM-ResNet) using erythrocyte microscopic image for malaria detection

			✓	Accuracy of 98.08 % slightly low compared to GoogleNet

Outperformed GoogleNet in term of error rate.

	
Wang et al. (2020)	Utilised pseudo-3D DenseNet for bacterial colonies detection and classification.

Use four time-consecutive frames in each time-lapse imaging.

Image of bacterial colonies captured after incubation in 23.5 h.

		✓	✓	Bacterial colonies >95% true detected within 12 h.

Precision (99.2–100%).

Bacterial colonies classified at 80%.

	
Li et al. (2022)	Utilised a lens-free imaging modality using a TFT image sensor.

A time-lapse of CFU images were automatically collected within 5 min on chromogenic agar plates.

Detect and count colonies (E. coli and other coliforms) using two DNNs (DenseNet)

		✓	✓	CFU detection: 92.6% sensitivity 100% specificity

CFU classification: 100% accuracy

	
Beznik et al. (2022)	Utilised U-Net architecture for multiple bacterial colonies segmentation and automated counting.

Regular U-Net
U-Net with a pre-trained encoder from ResNet-152		✓		Detection of true positive of class Virulent (bvg+) and Avirulent (bvg−):

Both scores were less than 74.5% Counting (Precision)

Bgv+ (90.6%)
Bgv− (91.9%)	
Zhang et al. (2022)	Use surface-enhanced Raman spectroscopies (SERS) to detect three types of Salmonella bacterial colonies.

	✓			Stacking-CNN outperform all other models:

Salmonella Typhimurium (98.31%),
Salmonella Enteritidis (98.41%)
Salmonella Paratyphoid (97.89%)	

Challenges in cell analysis approaches

Advanced technology exists to fulfil the current requirements with no intention of replacing the medical expert. Instead, the technology assists medical experts in making disease screening and diagnoses toward improving patient management. In general, most cell analyses still rely on manual methods performed by experts. However, conventional manual handling involves more handiwork with all the laboratory procedures that need to be well-organised and scrutinised for the lifetime of cell culture. Besides, the manual analysis performance could lead to error-prone, and it is very time-consuming. The researchers have tried to conduct several experimental works and research in developing an analysis system or application for cell and colony detection, classification, and counting. Cell analysis has monumental purposes in assisting for accurate and fast results, which explains the rationale that the advanced technology is needed. Considering that the Coulter counter implementation can provide excellent accuracies and is less time-consuming and reliable, it has been used in countless laboratory applications by average medical technology (Brecher, Schneiderman & Williams, 1956). In 1969, the Coulter counter (Model F) was applied in a cell suspension experiment to count red blood cells in mice’s spleen colonies (Worton, Mcculloch & Till, 1969). This principle also became an industry foundation (Graham, 2003). Presently, Coulter Principle has been practically referred to as incorporating numerous numbers of automated cell counters. One of them is the Beckman Coulter instrument.

On the other hand, stem cell analysis is used for regenerative medicine and transplant, especially HSPC. CFU is crucially used to test in vitro functional assay of the haematopoietic system for HSPC differentiation of blood cell types into red blood cells, white blood cells, and platelet (Laurenti & Göttgens, 2018; Boyer et al., 2019). A standardisation of CFU assay has been evaluated in a previous report due to the poor reproducibility of the assay (Pamphilon et al., 2013). The Biomedical Excellent for Safer Transfusion (BEST) collaborative cellular therapy team initiates a survey to improve the CFU assay’s reproducibility. According to their surveys, most of the laboratories respond to CFU assays incubated for 14–16 days with a performance of validation studies and consistency of inter-laboratory procedures. A semi-automatic device for the standardised quantification validation method of CFUs HSC has been conducted to accurately characterise the stem cell (Velier et al., 2019). Their research focuses on validating standard methods using the most designated computer software with an instrument developed by STEMCELL Technologies with the trademark STEMvision™. The experimental work of the manual method was compared with STEMvision™ on quantification counting of CFU using 20 mobilised apheresis, 17 samples from bone marrow, and 13 samples from cord blood. The result highlights that STEMvision™ meets high standard criteria than the manual method by reducing variance coefficients of recurring, varying inter-operator, and high capability in characterising the CFU HSC products.

The CCD digital camera was implemented in microscopic image acquisition with a high-quality sensor used in a camcorder as it consists of millions of micro solar cells in a 2D array, each of which converted the light into electrons. CCD has been used widely in several experimental works for image acquisition as it can generate a high-quality image without misinterpreting in charge transportation over the chips. For example, a CCD camera has been used in stem cell-derived extracellular vesicles for super-resolution microscopic image acquisition for MSCs analysis (Nizamudeen et al., 2018). The automated counting of MSCs by implementing a deep learning approach using microscopic images was acquired using a CCD digital camera and an inverted optical microscope (Hund Wetzlar) (Hassanlou, Meshgini & Alizadeh, 2019). In 2019, a digital image of bone marrow cells was acquired using a CCD camera of 20 megapixels (MP) on a single microscope (Yu et al., 2019). This research automatically identified and classified 11 classes from bone marrow cells using DNN, with the best achievement performed by erythroid lineage cells with 93.5% precision and 97.1% recall. Ramji et al. (2020) have applied the CCD sensors to correlate with one pixel colour in high-resolution image requirements. Unlike conventional sensors such as CMOS, the chips are manufactured traditionally like the standard microprocessors. Besides, CMOS tends to generate low image resolution vulnerable to image noise.

Research gaps can be highlighted for automated system development in the review of cell analysis approaches from the 16th century until the current approaches. Hitherto, the invention of microscopes in minuscule observation has inspired scientists and researchers to develop a physical instrument system for cell analysis based on photoelectronic and cell sorter techniques. The achievement of instruments involves uncountable repeated experimental works using laboratory apparatus incorporating electric circuits and an electronic device such as an oscillator. The implementation took much longer working hours than expected to be carried out of uncountable repeated experimental works and requires a high understanding of the multi-disciplinary work of cytological and engineering expertise. The efforts of previous scientists and researchers have motivated current researchers to carry out such trial and errors experimental works of simulation to develop a system without the need for laboratory experiment tools. All these opportunities could be implemented using digital visual sources such as images and videos using image processing approach. However, the implementation of DIP requirements is laborious and time constraints which involves multiple algorithms to select any significant input feature for the final decision system, such as cell morphology features. Most of the DIP and machine learning systems for cell analysation have semi-automated features for the system requires an update for additional features.

Therefore, researchers realised the rationale behind the need for a system that automatically updates the additional features without manual input, as the system can learn all possible features through deep learning. Nevertheless, the requirements of any digital camera tools are deniable for developing the automated system using the DL approach. The proposed automated system from the reviewed section in “Automated cell analysis using deep learning” can be referred to as a high potential framework point of view for future automated cell analysis systems regardless of the variety of microscopic images. In fact, despite the large-scale data requirements, the deep learning approach can reduce or eliminate manual feature extraction requirements and achieve a high-performance system in cell analysis.

Conclusions

This review studies have briefly discussed cell analysis which covers identification and counting evolution by time using traditional, conventional, and automated approaches. Various cell analysis has been proposed and performed by scientists and researchers to develop better cell analysis techniques for the automated system. All these initiatives were rationally introduced with the priority objective to assist the experts/laboratories in automatically identifying and counting the cell without needing handcrafted analysis. It is also to meet the current high demand for system automation applications. Advanced cell analysis has evolved much faster than traditional analysis with high performance as its high demand in technology. Therefore, this field has vast potential, particularly for industrial applications, despite a significant data requirement. A more automated system will be developed in the future, and cell analysis approaches will be innumerable.

Supplemental Information

Supplemental Information 1 PRISMA checklist.

Click here for additional data file.

Supplemental Information 2 Kindly refer to the table of corrections for the modification made to the revised manuscript.

Click here for additional data file.

Additional Information and Declarations

Competing Interests

Author Contributions

Data Availability

The authors declare that they have no competing interests.

Nurul Syahira Mohamad Zamani conceived and designed the experiments, performed the experiments, analyzed the data, prepared figures and/or tables, authored or reviewed drafts of the article, and approved the final draft.

Wan Mimi Diyana Wan Zaki conceived and designed the experiments, performed the experiments, analyzed the data, prepared figures and/or tables, authored or reviewed drafts of the article, and approved the final draft.

Zariyantey Abd Hamid analyzed the data, authored or reviewed drafts of the article, and approved the final draft.

Aqilah Baseri Huddin analyzed the data, prepared figures and/or tables, and approved the final draft.

The following information was supplied regarding data availability:

This article is a literature review.

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
