# Peer review of "Future stem cell analysis: progress and challenges towards state-of-the art approaches in automated cells analysis"

_PeerJ, doi:10.7717/peerj.14513_

## Round 0.1 · original submission · Minor Revisions

After the first round of review, the expert reviewers have suggested some critical minor revisions. I would like the authors to revise their manuscript to these valuable comments.

Reviewer 1 ·

Basic reporting

The first paragraph in the introduction seems to be mostly unnecessary. This manuscript is about stem cell analysis. Too much introduction on cells, in general, is not needed.

Thank you for providing a detailed review of cell analysis since the 16th century. I suggest focusing on the more recent technology. For example, the content from line 261 would be more of interest to the reader of this journal.

Same for the cell analysis using DIP and machine learning approaches, I recommend focusing on the content from line 357, which discusses more recent development.

The automated cell analysis using deep learning is the most interesting part of this manuscript. I recommend staying focused. For example, from lines 401-404, the story about Alan Turing and artificial intelligence is not relevant to the theme of this paper.

Experimental design

The literature research method used in this manuscript is thorough. The related studies were identified and summarized.

Validity of the findings

Thank you for providing the description of the survey methodology. The method is described in detail.

Reviewer 2 ·

Basic reporting

- This present manuscript is a broad-stroked review of the field of stem cell analysis using modern machine learning and image processing techniques. As such, it is a well-written article with cross-disciplinary interests.
- While I am aware of reviews on ML techniques for cell analysis, I am unaware of an article quite as comprehensive as the present article. I have, however, suggested several ways in which readability of this article can be improved.
- This article is approachable to readers with varied backgrounds, and I think that is a strong prerequisite for a good review article. I congratulate the authors for producing this manuscript.

Experimental design

- Clear and crisp usage of English throughout the text. I congratulate the authors for producing this manuscript. The section on survey methodology clearly indicates the body of reference papers that the present survey analyses and summarizes – which, by itself, is very thorough.
- The sections on overview and evolution of stem cell research does very good justice to reviewing most relevant work, starting from early 2000s to 2020. However, I recommend that the summary of older work (lines 182-336) be shortened to no more than 2 paragraphs.
- In the section on DIP and ML approaches, the authors again mention work from a long time ago – I would advise removing them (lines 345-356). Also, only a couple of lines are dedicated to ML techniques (and only SVM, at that). I recommend either expanding to review more work on ML (not DL) or coalesce this with the section on deep learning techniques.
- In lines 399-405, I think this can be safely removed without the risk of incompleteness for this review. I also wonder if the performance results of this section can instead be summarized in a table – it is indeed difficult to follow the results across various techniques that have been reviewed. Say, the columns are: [year, technique/algorithm, model parameters, results, reference]. It would be easier for the readers to follow the evolution of cell analysis using ML.

Validity of the findings

- The present article concludes with challenges in cell analysis approaches. While this section is well-written, I would also like to see some commentary on future directions.
- There has been a lot of work on image processing that has come up over the past year or so. However, the work punctuates references on ML until 2020 – I might suggest reviewing any work that uses more recent ML work; this however is not mandatory.

Additional comments

None, apart from the ones above.

---

## Round 0.2 · accepted · Accept

The authors successfully implemented the reviewers' comments. The manuscript can now be acceptable as it is. I would like to congratulate the authors once again for their work.

Reviewer 1 ·

Basic reporting

The author has resolved all comments from the first round of review. The author has rearranged the section and focused more on recent technologies for cell analysis. The same applies to the DIP and machine learning approaches. More recent development has been discussed. The author has removed the story about Alan Turing and artificial intelligence which is not relevant to the theme of this paper.

Experimental design

Thank you for providing the description of the survey methodology. The method is described in detail. The articles cited in this manuscript have been quoted in other studies as well. The first paragraph was unnecessary but the author has made the change to improve it.

Validity of the findings

The author has added discussion about future directions to the manuscript.

Additional comments

None, apart from the ones above.

Reviewer 2 ·

Basic reporting

The authors have shorted the history of stem cell analysis - I am satisfied with the length of the article as it stands. As I remarked previously, this article is approachable to readers with varied backgrounds, and I think that is a strong prerequisite for a good review article. I congratulate the authors for producing this manuscript.

Experimental design

- The authors have cut down a lot of extraneous sentences that were not helping with the logical flow of the article.
- They have also added more references to background articles, which is all the more helpful. I am especially happy about the inclusion of Table 1 and Table 2, which very comprehensively describe the present state of the art in stem cell research. I thank the authors for considering this suggestion.
- Between lines 260 and 274, the authors have added reference to more work on cell analysis using ML - I am glad about this,.

Validity of the findings

No new comments here.

Additional comments

I commend the authors for this article, and I support the article's publication.

---

## Author Rebuttal · Round 0.2

**Fakulti Kejuruteraan dan Alam Bina**    *Faculty of Engineering and Built Environment*

# UNIVERSITI KEBANGSAAN MALAYSIA
*The National University of Malaysia*

**Date: 03 November 2022**

**Original Article Title: "Future Stem Cell Analysis: Progress and Challenges Towards State-of-the art Approaches in Automated Cells Analysis"**

**To:** PeerJ Journal Editor

**Re:** Response to reviewers

Dear Editor,

Manuscript Resubmission (**Original Manuscript ID:** 2022:09:77216:0)

Thank you very much for your consideration of our manuscript and for allowing a resubmission of our manuscript, with very positive valuable comments and the opportunity to address the reviewers' comments. We have copied and pasted all reviewers' comments and addressed each one individually.

As you will see, we have made every attempt to incorporate the suggestions made by all the reviewers and have inserted them into the revised manuscript where appropriate thoroughly. We hope our revision has improved the paper to the level of their satisfaction and the editors. Number-wise answers to their specific comments/suggestions/queries are as attached.

Thank you.

Yours faithfully,

(Wan Mimi Diyana Wan Zaki, PhD)

**FAKULTI KEJURUTERAAN DAN ALAM BINA, *FACULTY OF ENGINEERING AND BUILT ENVIRONMENT***
Universiti Kebangsaan Malaysia, 43600 UKM Bangi, Selangor Darul Ehsan, Malaysia
Telefon: +603-8927 2462   Faksimili: +603-8921 6452   E-mel: wmdiyana@ukm.edu.my

Mengilham Harapan, Mencipta Masa Depan • *Inspiring Futures, Nurturing Possibilities*    www.ukm.my

**Responses to Reviewer**
**(Manuscript ID:** 2022:09:77216:0**)**

# REVIEWER 1

**Basic reporting:**

1. The first paragraph in the introduction seems to be mostly unnecessary. This manuscript is about stem cell analysis. Too much introduction on cells, in general, is not needed.

**Response:**
Thank you for your feedback. We have rephrased the first paragraph and removed unnecessary introduction on cells as mentioned. Kindly refer to line 37 to line 43.

2. Thank you for providing a detailed review of cell analysis since the 16th century. I suggest focusing on the more recent technology. For example, the content from line 261 would be more of interest to the reader of this journal.

**Response:**
Thank you very much for your feedback. We apologized for the lengthy arrangement of the Section: The history of cell analysis (from 16th century) in the manuscript. Therefore, we rearranged and revised the section which focused on the more recent technology. We believe that this new arrangement has improved the overall clarity of the manuscript, and we hope that it is in accordance with the reviewer's suggestions. Considering your feedback & Reviewer 2 comments, the older summary of work (previously in lines 182-336) has been shortened so that we are focusing on the more recent technology. Kindly refer to line 165 to line 214.

3. Same for the cell analysis using DIP and machine learning approaches, I recommend focusing on the content from line 357, which discusses more recent development.

**Response:**
Considering your feedback and Reviewer 2's to stay focus on the recent technology, we have made the amendments as follows:

| Previous manuscript | Referred lines/Figure |
| --- | --- |
| We have removed the older work on the invention of the camera in Line 345-356 so that we are focusing on the content from | Kindly refer to the revised paragraph from line 217 to line 223 |

**FAKULTI KEJURUTERAAN DAN ALAM BINA,** *FACULTY OF ENGINEERING AND BUILT ENVIRONMENT*
Universiti Kebangsaan Malaysia, 43600 UKM Bangi, Selangor Darul Ehsan, Malaysia
Telefon: +603-8927 2462   Faksimili: +603-8921 6452   E-mel: wmdiyana@ukm.edu.my

| line 357 as recommended, currently in line 224. | |
| --- | --- |
| One paragraph added to provide a better clarity of the recent ML work on the cell analysis. | Kindly refer to line 260 to line 274. |
| Three additional recent ML works are added in Reference section. | Kindly refer to,<br>Additional reference 1: Line 686-688<br>Additional reference 2: Line 728-730<br>Additional reference 3: Line 773-776 |
| The number of total articles referred is therefore updated. | Kindly refer to Line 102, Line 106, and Line 107 in section Survey Methodology. |
| We have improved the Figure 1 and Figure 2 for the changes made. | Kindly refer to:<br>Figure 1-first decision attachment.<br>Figure 2-first decision attachment |

4. The automated cell analysis using deep learning is the most interesting part of this manuscript. I recommend staying focused. For example, from lines 401-404, the story about Alan Turing and artificial intelligence is not relevant to the theme of this paper.

**Response:**
Thank you for your positive comments. We have removed the story about Alan Turing and AI as suggested.

**Experimental design:**

5. The literature research method used in this manuscript is thorough. The related studies were identified and summarized.

**Response:**
Thank you for your positive remarks.

**Validity of the findings:**

6. Thank you for providing the description of the survey methodology. The method is described in detail.

**Response:**
Thank you for your positive remarks.

## REVIEWER 2

**Basic reporting:**

1. This present manuscript is a broad-stroked review of the field of stem cell analysis using modern machine learning and image processing techniques. As such, it is a well-written article with cross-disciplinary interests. While I am aware of reviews on ML techniques for cell analysis, I am unaware of an article quite as comprehensive as the present article. I have, however, suggested several ways in which readability of this article can be improved. This article is approachable to readers with varied backgrounds, and I think that is a strong prerequisite for a good review article. I congratulate the authors for producing this manuscript.

**Response:**
Thank you for your positive remarks.

**Experimental design:**

1. Clear and crisp usage of English throughout the text. I congratulate the authors for producing this manuscript. The section on survey methodology clearly indicates the body of reference papers that the present survey analyses and summarizes – which, by itself, is very thorough.

**Response:**
Thank you very much for your positive and motivating comments.

2. The sections on overview and evolution of stem cell research does very good justice to reviewing most relevant work, starting from early 2000s to 2020. However, I recommend that the summary of older work (lines 182-336) be shortened to no more than 2 paragraphs.

**Response:**
Thank you very much for your feedback. We apologized for the lengthy arrangement of the Section: The history of cell analysis (from 16th century) in the manuscript. Therefore, we rearranged and revised the section which focused on the more recent technology. We believe that this new arrangement has improved the overall clarity of the manuscript, and we hope that it is in accordance with the reviewer's suggestions. Considering your feedback & Reviewer 1 comments, the older the summary of older

work (previously in lines 182-336) has been shortened so that we are focusing on the more recent technology. Kindly refer to line 165 to line 214.

3. In the section on DIP and ML approaches, the authors again mention work from a long time ago – I would advise removing them (lines 345-356). Also, only a couple of lines are dedicated to ML techniques (and only SVM, at that). I recommend either expanding to review more work on ML (not DL) or coalesce this with the section on deep learning techniques.

**Response:**
We have removed the older work on the invention of the camera in Line 345-356 as suggested. Considering your feedback and Reviewer 1's to stay focused on the recent technology, we have made the amendments as follows:

| Previous manuscript | Referred lines/Figure |
| --- | --- |
| We have removed the older work on the invention of the camera in Line 345-356 so that we are focusing on the content from line 357 as recommended, currently in line 224. | Kindly refer to the revised paragraph from line 217 to line 223 |
| One paragraph added to provide a better clarity of the recent ML work on the cell analysis. | Kindly refer to line 260 to line 274. |
| Three additional recent ML works are added in Reference section. | Kindly refer to, Additional reference 1: Line 686-688 Additional reference 2: Line 728-730 Additional reference 3: Line 773-776 |
| The number of total articles referred is therefore updated. | Kindly refer to Line 102, Line 106, and Line 107 in section Survey Methodology. |
| We have improved the Figure 1 and Figure 2 for the changes made. | Kindly refer to: Figure 1-first decision attachment. Figure 2-first decision attachment |

4. In lines 399-405, I think this can be safely removed without the risk of incompleteness for this review. I also wonder if the performance results of this section can instead be summarized in a table – it is indeed difficult to follow the results across various techniques that have been reviewed. Say, the columns are: [year, technique/algorithm, model parameters, results, reference]. It would be easier for the readers to follow the evolution of cell analysis using ML

**Response:**
Thank you for your comments. We have removed lines 399-405 as suggested. I would like to re-inform; we have already summarised this reviewed section in Table 2. Kindly refer to the Table 2 attachment for the evolution of cell analysis using DL.

**Validity of the findings:**

5. The present article concludes with challenges in cell analysis approaches. While this section is well-written, I would also like to see some commentary on future directions.

**Response:**
Thank you for your positive comments. We have revised the paragraph and added some commentary on future direction as suggested. Kindly refer to line 491 to line 499.

6. There has been a lot of work on image processing that has come up over the past year or so. However, the work punctuates references on ML until 2020 – I might suggest reviewing any work that uses more recent ML work; this however is not mandatory.

**Response:**
Thank you very much for your comment. To the best of our knowledge, we have considered and made the amendments for the recent work on cell analysis using ML in the manuscript as mentioned in Comment 3. Kindly refer to line 260 to line 274 for one additional paragraph of the recent work on cell analysis using ML.

*Additional comments:*

7. None, apart from the ones above.

**Response:**
Thank you for your comment.